# Mobilization of Children with External Ventricular Drains: A Retrospective Cohort Study

**DOI:** 10.3390/children9111777

**Published:** 2022-11-19

**Authors:** Ben Reader, Emily Stegeman, Nanhua Zhang, Kelly Greve

**Affiliations:** 1Division of Clinical Therapies, Nationwide Children’s Hospital, Columbus, OH 43205, USA; 2Division of Occupational Therapy and Physical Therapy, Cincinnati Children’s Hospital Medical Center, Cincinnati, OH 45229, USA; 3Division of Biostatistics and Epidemiology, Cincinnati Children’s Hospital Medical Center, Cincinnati, OH 45229, USA; 4College of Medicine, University of Cincinnati, Cincinnati, OH 45221, USA; 5Department of Rehabilitation, Exercise and Nutrition Sciences, College of Allied Health Sciences, University of Cincinnati, Cincinnati, OH 45221, USA

**Keywords:** physical therapy, external ventricular drain, rehabilitation, neurosurgery, pediatrics

## Abstract

The implementation of early mobility programs for children with critical illnesses has been growing. Children with acute neurologic conditions that result in the requirement of an external ventricular drain (EVD) may be excluded from attaining the benefits of early mobility programs due to the fear of adverse events. The purpose of this study was to examine the implementation, safety, and outcomes of children with EVDs mobilized by physical therapists. A single-site retrospective cohort study of children with EVDs mobilized by physical therapy (PT) was conducted. Patients aged 3–21 years who were hospitalized from September 2016 to December 2020 were included in this study. Results: Out of a total of 192 electronic health records with EVDs, 168 patients (87.5%) participated in 1601 early mobilization encounters led by physical therapists. No adverse events occurred due to mobilization. Patients mobilized more frequently by PT had a higher level of activity at discharge (*p* = 0.014), a shorter length of stay (*p* = 0.001), and a more favorable discharge (*p* = 0.03). The early mobilization of children with EVDs can be implemented safely without adverse events. Patients mobilized with an EVD are more functional at discharge, spend fewer days in the hospital, and have a more favorable discharge compared to those who do not receive PT.

## 1. Introduction

External Ventricular Drains (EVD), also known as extraventricular drains or ventriculostomy, are a commonly utilized form of fluid drainage in patients with neurological deficits to remove excess cerebrospinal fluid (CSF) or intraventricular blood, normalize and measure intracranial pressure (ICP), and allow for the adequate healing of the cerebrum [1,2,3]. In pediatric populations, the most common diagnoses requiring the placement of an EVD include traumatic brain injury, ventriculoperitoneal shunt failure, and new-onset hydrocephalus [4]. The importance of the therapeutic drainage of CSF or intraventricular blood in the prevention of brain injury makes EVDs critically important lines. Due to the intrusive nature and precarious placement of EVDs, great care and attention must be paid to ensure the integrity of the catheter and all other components [2].

Individuals with EVDs on neurosurgical or intensive care units are often placed on bedrest due to fear of the adverse effects of mobilization [5]. Potential adverse effects of mobilization include: (1) dislodgement of the EVD catheter, (2) exacerbation of delayed cerebral ischemia, (3) increased ICP, (4) falls, (5) over-drainage of CSF, and (6) permanent complications [6]. In addition to the fragility of the securement of the EVD, considerations that may impact the safety and feasibility of the mobilization of patients with EVDs are the potential impulsivity and the high risk of falls as a result of cerebral damage [7].

Immobility and prolonged bedrest is known to induce sarcopenia, elevate the risk of thrombotic events, and increase the likelihood of death [8,9]. Following critical illness, children demonstrate a greater risk of muscular weakness, poorer functional ability, a reduced quality of life, and increased healthcare expenditure [10,11]. To prevent the onset of the negative sequalae of events that are common in post-critical illness, early mobilization programs have been implemented safely and demonstrate positive outcomes [12,13,14,15,16].

In the adult literature, there is a growing body of evidence suggesting that the early mobilization of individuals with EVDs is safe, feasible, and has no long-lasting adverse effects [5,7,17,18,19,20], despite the unique challenges posed by patients in neurocritical states [21]. To prevent the onset of the negative sequelae of prolonged hospitalization, the calculated risk of early mobilization under the guidance of physical therapists has demonstrated improved discharge dispositions, a reduced length of stay, and a reduced need for mechanical ventilation [6]. The mobilization of children with EVDs is common practice in large pediatric institutions and is reported to be a precaution rather than a contraindication [15,22,23], though, to the authors’ knowledge, there is no existing literature to support the mobilization of children with EVDs. Associated risks in the pediatric setting may include escalating and challenging behaviors. Children can sometimes argue, show aggression, or act angry or defiant around others when they are anxious or stressed. A hospital can be a stressful environment and can lead to challenging behaviors inclusive of the physical behaviors of pushing/pulling/biting/hitting/kicking, pulling at lines, defiant behaviors, active flight risk, or variable emotions of screaming, yelling, and crying. Children admitted for medical care can often show resistance to medical treatment and necessary care due to past healthcare experiences, temperament, sensory concerns or sensory processing disorders, reactive behaviors due to stressors and pain, mental health diagnoses, and altered mental statuses. Children exhibit the same negative sequelae as adults from prolonged immobility; therefore, there is a need to assess the safety and outcomes of mobilizing children with EVDs.

At Cincinnati Children’s Hospital, the medical team provides a nursing and therapy protocol for the mobilization for children with an EVD. (Figure 1) The protocol for nursing to clamp and unclamp the EVD is aligned with the process outlined by Muralidharan [2]. PT consultation is provided if concerns regarding functional deficits arise. When referred, the PT consultation provides guidance regarding out-of-bed activity, a 15-to-30 min allotted time for the EVD to be clamped, and activity precautions. Mobilization with PT only begins once nursing has clamped the EVD. While the EVD is clamped, the patient is allowed to complete therapeutic activities to progress in terms of strength, balance, and coordination in an attempt to return to baseline gross motor activities. Two staff members are present for out-of-bed activity: one to assist the patient and the other to assist with EVD hardware management during mobility. Following the allotted clamp time, the patient is returned to the bed or positioned safely in a chair, where the drain is unclamped and leveled by nursing staff.

Rapid response team (RRT) activation occurs with medical instability, the criteria for which include increased work of breathing, agitation/decreased consciousness, or staff/parental concern [24]. The activation of the RRT would occur if adverse events from mobilization are presented or if patients become medically unstable.

The purpose of this study was to explore the implementation, safety, and outcomes of children with EVDs mobilized while admitted at a large, academic pediatric hospital. It was hypothesized that children with EVDs could participate in therapeutic mobilization without adverse effects and demonstrate improved functional outcomes.

## 2. Materials and Methods

### 2.1. Study Design

A retrospective cohort study was completed using the electronic health records (EHR) of children and adolescents admitted to a neurology inpatient unit at Cincinnati Children’s Hospital with current EVDs from September 2016 to December 2020. Children were included if they were between the ages of 3 and 21 years and had at least one EVD during an inpatient stay. Exclusion criteria included children who were under 3 years of age. Approval for this study was granted by the hospital’s Institutional Review Board.

### 2.2. Data Extraction

Clinical data were extracted through the EHR. Data were manually sorted and cleaned by two researchers [BR, ES]. The extracted variables included age, gender, medical diagnosis, number of EVDs, presence of physical therapy (PT) orders, number of days elapsed between EVD placement and PT evaluation, presence of bedrest orders, RRT activation, and PT frequency. The dependent variables that were extracted included the level of mobility present at PT evaluation and achieved by discharge, length of admission, and discharge disposition. To assess safety, PT and RRT notes were examined to determine whether mobilization was the cause of, or contributed to, medical instability. If RRT was linked to mobilization, it was reported as an adverse event. The presence of RRT notes that were deemed unrelated to mobilization were utilized as a measure of medical instability.

Activities included in the umbrella term of early mobilization in pediatric settings have not been clearly established. Activities specifically categorized into ranked difficulty level included range of motion (ROM), bed mobility, sitting, standing, ambulation with assistance, ambulation without assistance, and higher-level balance activities (e.g., single leg stance, cone drills, uneven surfaces). From the extracted data, level of activity achieved at evaluation and discharge was manually categorized into one of the above categories and agreed upon by two researchers [BR, ES]. The PT-related discharge disposition of the patient was categorized manually into home, outpatient/home-health PT, acute/sub-acute rehabilitation, or hospice/death.

### 2.3. Statistical Analysis

Demographic and baseline characteristics were summarized using means (standard deviation [SD]) or medians (interquartile range [IQR]) for continuous variables and proportions for discrete variables. Two-sample t-tests and Wilcoxon and Chi-square tests were used to compare those who received PT evaluations and those who did not. Among those who received PT evaluations, comparisons were made between those who were recommended for PT at different frequencies (<1x/week, 1–2x/week, 3–5x/week, 6x/week) in terms of demographic and baseline characteristics.

The main objective of the analysis was to study the effect of PT on the level of mobility achieved at evaluation and discharge, length of stay, and discharge disposition. We also examined the association between patient conditions and these outcomes, and these conditions included diagnosis, recommended PT frequencies, bedrest orders, and days between admission and PT evaluation. Cumulative logistic regression models were constructed for the level of mobility achieved at discharge and discharge disposition; a general linear model with log link was used to model the length of stay. PT evaluation status was the primary explanatory variable, and confounders were controlled in all regression models. Among those who received PT, separate models were used to examine whether the recommended PT frequency was associated with the outcomes. As a secondary analysis, the association between demographic variables and bedrest order was examined using Chi-square tests for discrete demographic data and the Mann–Whitney U tests for continuous demographic variables. Statistical analysis was performed using JASP for Mac, Version 0.14 (University of Amsterdam, The Netherlands) and SAS version 9.4 (SAS Institute, Cary, NC, USA).

## 3. Results

A total of 192 EHRs met the inclusion criteria for this study, with 168 (87.5%) patients receiving a PT evaluation. A total of 1601 patient PT encounters were analyzed. Fifty-one patients (26.6%) demonstrated medical instability and required RRT activation that was unrelated to mobilization. No adverse events of mobilization occurred during any of the 1601 patient PT encounters.

### 3.1. Patient Characteristics:

Demographics and patient characteristics can be found in Table 1 and Table 2, respectively. The patients were a mean age of 9.93 years (SD 4.91). Male patients outnumbered female patients, but no association was found between genders in any category. The number of EVDs per patient ranged between 1 and 3, though no significance between variables was found for the number of EVDs per patient and the activity at evaluation, activity at discharge, or discharge disposition.

### 3.2. Activity at Evaluation

Diagnosis significantly impacted the activity level at PT evaluation (*p* = 0.01). Patients with neoplasms were more likely to achieve a higher level of activity at evaluation. Patients with intracranial hemorrhage (ICH) or infections were more likely to have a lower activity at evaluation. Children with a lower level of activity achieved at evaluation had an overall lower recommended PT frequency (*p* < 0.0001).

### 3.3. Activity at Discharge

Patients with a diagnosis of ICH or infection had 4.44 and 3.77 times the odds of having a higher level of mobility at discharge, respectively, compared to those with a shunt malfunction. Patients with bedrest orders were significantly more likely to have a decreased level of activity at discharge compared to the activity level achieved by patients without bedrest orders (*p* < 0.001). The mean number of days to PT evaluation was significantly longer among patients sitting (7.95, 9.30 SD) compared to ambulation with assistance (3.89, 2.43 SD), ambulation without assistance (3.48, 2.74 SD), and higher-level balance activities (3.2, 1.58 SD) at discharge. The cumulative logistic regression model revealed that patients with a higher recommended PT frequency were significantly associated with a higher level of activity at discharge (*p* = 0.014) after controlling for confounding factors, although post hoc pairwise comparison did not find any significant pairwise difference.

### 3.4. Length of Stay

Children who had a diagnosis of ICH, infection, neoplasm, or subdural hematoma (SDH) had a higher expected length of stay compared to those who had shunt malfunction (*p* < 0.001), while patients who had a functional hemispherectomy diagnosis had a shorter expected length of stay compared to children with shunt malfunction (*p* < 0.001). The number of EVDs per patient significantly impacted the length of stay (*p* = 0.02). Patients with one EVD spent fewer days in the hospital (median = 21 days, IQR 26.50) compared to children with two EVDs (median = 49 days, IQR 40.75) and three EVDs (median = 67 days, IQR 54.00). The multivariate Poisson regression model showed that the presence of a PT evaluation was significantly associated with an increased length of stay (*p* < 0.0001) after controlling for confounders; the log of the expected length of stay was 0.66 (95% C.I. 0.54–0.78) higher among patients who received a PT evaluation compared to those who did not. The greater the number of days between admission and PT evaluation, the longer the expected length of stay (*p* < 0.0001). A lower PT frequency equated to a longer length of stay (*p* = 0.001).

### 3.5. Discharge Disposition

Patients who had an ICH diagnosis had lower odds of having a less desirable discharge disposition compared with patients with other diagnoses. Older age was associated with less desirable discharge disposition outcomes. Analysis showed that the presence of bedrest orders significantly increased the likelihood of less desirable discharge disposition compared to those without bedrest orders (*p* < 0.001). The number of days to PT evaluation did not affect discharge disposition (*p* = 0.44). The cumulative logistic regression of the discharge disposition indicated that the recommended PT frequency was significantly associated with discharge disposition (*p* = 0.03); children with less than 6x/week of recommended PT frequency had at least 2.44 times the odds of a less desirable discharge disposition compared to those who were recommended 6x/week.

### 3.6. Medical Instability

Children who received PT evaluation were more likely to have RRT activation compared to those who did not receive PT evaluation (30.36% vs. 0%, *p* = 0.002). Among patients who received PT evaluations, children with documented RRT activation were more likely to have a less recommended frequency of PT (*p* = 0.083). Those who did not have documented RRT activation had 2.08 times the odds of having a more favorable discharge disposition.

## 4. Discussion

To the authors’ knowledge, this is the first examination of clinical data assessing the implementation, safety, and outcomes achieved by mobilizing pediatric patients with EVDs. The findings of this retrospective cohort study demonstrate that most children with at least one EVD received PT-led mobilization, and there were no incidents of adverse events. Children who were evaluated by PT soon after EVD placement and seen regularly for PT intervention demonstrated increased functional mobility at discharge and a shorter length of stay in the hospital.

### 4.1. Implementation

The early mobilization of critically ill adult patients has received a high-level research interest and has recently been gaining attention in pediatric populations [15,16,23,25]. Despite the increased attention in the literature, barriers to the mobilization of children, such as patients with EVDs, are still commonly cited [22]. Others have reported clinician education as a barrier to the mobilization of patients with EVDs and have suggested the need for best-practice guidelines to encourage participation [26]. Excluding patients with EVDs from early mobilization during the critically ill phase denies those children the ability to attain the improved outcomes that have been reported. Despite having EVDs, the majority (*n* = 168, 87.5%) of patients included in this study were able to perform a form of mobilization at the initial PT evaluation and in subsequent sessions. Definitions of mobilization activities vary between studies due to lack of standardization. One previous study included activities ranging from passive ROM to walking without assistance [19], and others have only included out-of-bed activities [5,7]. For this study, passive ROM to a higher level of activities, including stairs and balance training, were incorporated in all sessions to prevent the onset of negative impacts of immobility occurring as a result of critical illness. Others have explored the use of bed cycle ergometers to allow adults with EVDs [27] and children with critical illnesses [21] to gain the benefit of mobilization while on bedrest, though no studies have examined the safety and feasibility for children with EVDs.

### 4.2. Safety

The safe mobilization of adult patients with EVDs has been demonstrated to defy claims of safety being a barrier to participation [5,6,7,17,19,20]. Though similar barriers have been reported in the pediatric literature [22,23], our retrospective cohort study provides evidence that mobilization for pediatric patients can be performed safely and without adverse events. A progressive approach to mobility, standardized safety criteria, interdisciplinary collaboration, and the securement of lines/drains have all been cited as strategies to overcome barriers impacting the safety of critically ill patients [28]. The protocol in place at our institution meets this recommendation and allows for the safe mobilization of pediatric patients with EVDs. The aim of physical therapy intervention with an EVD is to prevent deconditioning, weakness, and immobility while keeping patients safe and protected during these critical and vulnerable states.

Patients with RRT activation and more medically unstable patients were seen less frequently by PT, likely due to limitations based on their clinical status—for example, children post-craniectomy are limited in their participation until a custom helmet can be provided. Patients who were more medically stable had greater odds of being able to return home following hospitalization. The impact of medical stability on discharge disposition has not been examined in previous literature, making comparisons impossible. Although not directly analyzed, child behavior did not appear to limit safety during mobilization in the cohort of children in our study.

### 4.3. Outcomes

Due to medical instability, 61% of children with ICH are admitted to the intensive care unit [17]. This medical instability immediately following ICH may explain the lower activity level at PT evaluation for those with ICH. Residual deficits and morbidity following ICH are common for pediatric patients and can lead to long-term deficits in physical functioning and social participation [29]. Additionally, children with ICH were more likely to require acute/subacute rehabilitation and hospice services compared to other diagnoses. Children with ventriculoperitoneal shunts have a pre-existing condition that necessitates the surgical insertion of a shunt, most commonly hydrocephalus [30]. Compared to patients with new onset diagnoses such as ICH, neoplasms, and infections, children with shunts are likely to present with long-standing functional deficits that could impact their rehabilitation potential. The cohort findings indicate that patients with shunt malfunctions had lower levels of activity at discharge compared to those with ICH and infections, but they were able to discharge sooner. It is possible that this is due to patients reaching their pre-admission baseline activity level, which may have been lower compared to those with other, new-onset diagnoses such as ICH, neoplasms, or infections.

The length of stay for children with PT orders in this cohort was, on average, 36 days, which is comparatively much higher than the 17–24 days reported in the adult literature [5,6,7]. A similar number of days to first mobilization post-EVD placement was present in this cohort (4.5 days) when compared to the 3.5–7.7 days reported in the adult literature [5,6,7,19,20]. In our sample, children who were mobilized earlier demonstrated higher levels of activity at discharge and spent fewer days in the hospital. It is possible that the early mobilization of patients was able to prevent the negative sequalae of hospital-acquired deconditioning. In addition to seeing the patients earlier, children with a higher frequency of PT intervention demonstrated improved outcomes such as higher activity levels at discharge, a shorter length of stay, and more favorable discharge dispositions.

Though out of the control of the physical therapist, the impact of bedrest orders resulted in poorer functional outcomes post-EVD placement. Activity at discharge was lower and discharge dispositions were less favorable for those with bedrest orders. Mobilization was limited for patients with bedrest orders to activities in the bed, reducing the ability of PT to implement functional, task-specific activities such as ambulation that have been shown to lead to improved outcomes [31].

### 4.4. Limitations

The limitations to this study include generalizability, data extraction, and comparable literature. The generalizability of this study is limited due to the specific protocol implemented at the authors’ facility that may not be available at other hospital settings. Due to the retrospective study, categories were manually chosen by two researchers rather than extracted from the EHR. The comparison of children who received PT to those who did not, based on diagnosis, was limited due to the lack of known outcomes in children who did not receive PT, as well as the comparison of existing research available in this patient population; hence, future studies should be performed to corroborate our findings. Relatively small sample sizes, specifically in the <1x/week frequency group, may not have adequately powered the statistical and post hoc testing that was performed as part of this study, which limits pairwise analysis. Patient age-specific risks, such as behavior, are likely to be important factors in rehabilitation outcomes, though they are difficult to measure and were not included in this study. This study excluded patients under 3 years of age and cannot be translated to younger populations; future work should aim to assess the benefits of early mobilization for pre-ambulatory patient populations.

## 5. Conclusions

The early mobilization of pediatric patients with EVDs can be implemented safely with proper safeguards in place. PT involvement results in better outcomes including improved functional activity levels at discharge, a reduced length of hospital admission, and more favorable discharge dispositions. Physical therapists, neurosurgeons, and nurses should advocate for early PT referral for the early mobilization of children with EVDs to allow for greater functional outcomes.

## Figures and Tables

**Figure 1 children-09-01777-f001:**
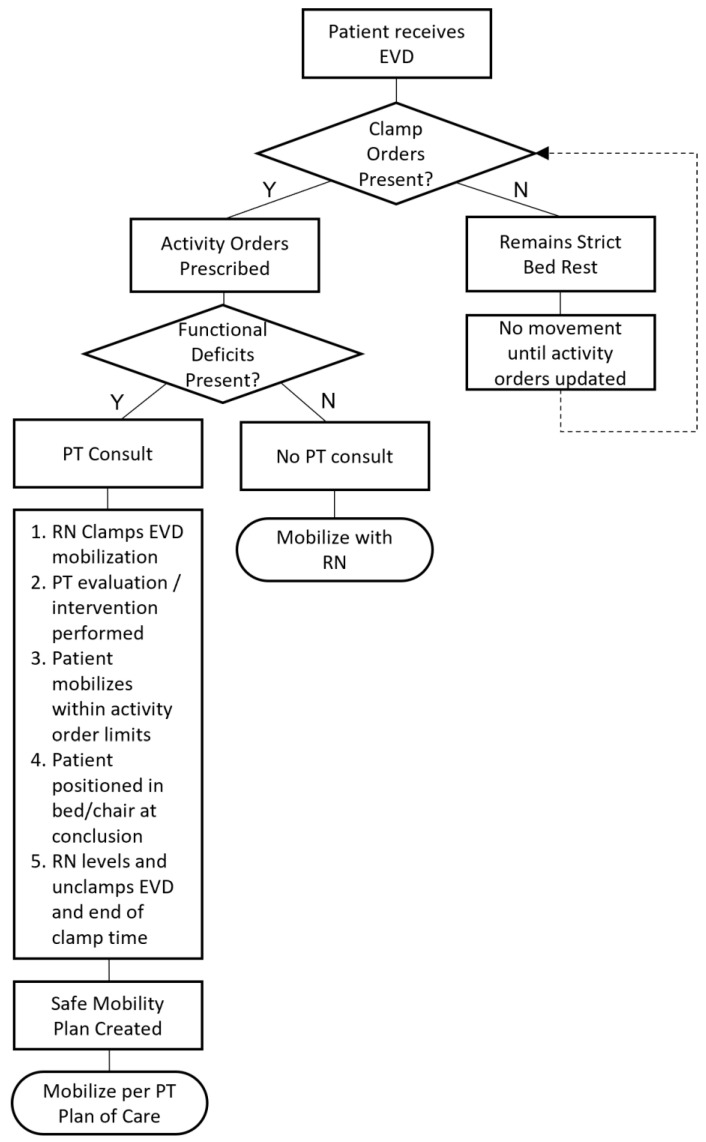
Registered Nurse (RN) and Therapy Protocol for the Mobilization for Children with an External Ventricular Drain (EVD).

**Table 1 children-09-01777-t001:** Demographics of Patients with External Ventricular Drains.

	Total *n* (%)	With PT Orders *n* (%)	Without PT Orders *n* (%)
Number of Records	192 (100)	168 (87.5)	24 (12.5)
Sex (*n*, %) Male Female	104 (54.2) 88 (45.8)	89 (53) 79 (47)	13 (54.2) 9 (45.8)
PT Encounters	1601 (100)	1601 (100)	N/A
Bedrest Orders	26 (15.5)	26 (15.5)	Unknown
Diagnosis ICH Infection Neoplasm SDH Shunt Malfunction FH Other	24 (12.5) 19 (9.9) 63 (32.8) 6 (3.1) 46 (24.0) 14 (7.3) 20 (10.4)	22 (13.1) 19 (11.3) 56 (33.3) 5 (3) 35 (20.8) 14 (8.3) 17 (10.1)	2 (8.3) 0 (0.0) 7 (29.2) 1 (4.2) 11 (45.8) 0 (0.0) 3 (12.5)
PT Frequency <1x/week 1–2x/week 3–5x/week 6x/week	6 (3.6) 48 (28.6) 78 (46.4) 36 (21.4)	6 (3.6) 48 (28.6) 78 (46.4) 36 (21.4)	N/A N/A N/A N/A

FH = Functional Hemispherectomy; ICH = Intracerebral Hemorrhage; PT = Physical Therapy; SDH = Subdural Hematoma.

**Table 2 children-09-01777-t002:** Mean Age, Length of Stay, Number of EVDs, Days to Evaluation.

	Total Mean ± SD	With PT Orders Mean ± SD	Without PT Orders Mean ± SD
Age (years)	9.9 ± 4.9	9.8 ± 4.8	11.0 ± 5.4
Length of Stay	33.4 ± 36.2	36.4 ± 38.2	12.7 ± 14.0
Number of EVDs	1.1 ± 0.3	1.1 ± 0.3	Unknown
Days to PT Evaluation	4.5 ± 4.4	4.5 ± 4.4	N/A

EVDs = External Ventricular Drains; PT = Physical Therapy.

## Data Availability

The data presented in this study are available on request from the corresponding author. The data are not publicly available due to size of data file.

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
