# Peer review of "Mobilization of Children with External Ventricular Drains: A Retrospective Cohort Study"

_children, 2022, doi:10.3390/children9111777_

Round 1

Reviewer 1 Report

no comment this article

Author Response

We are grateful for the time you took to provide feedback and your positive review of our manuscript. Thank you for your input.

Reviewer 2 Report

This is clearly designed investigation which shows safety of mobilization of a certain patients with EVD. 

Author Response

We are very appreciative for your supportive comments and favorable review. Thank you for taking time to review our manuscript.

Reviewer 3 Report

The present paper deals with the effects of early mobilization guided by physical therapists, in children with external ventricular drain placement. Despite growing literature evidence of its benefits in adults, due to preventing of the negative consequences of long-time resting, the mobilization of children with EVD is lacking literature data, and is connected with the increased fear of the adverse effects due to the fragility and complexity of the placement.

This retrospective study systematizes data on the PTs conducted in children with EVDs in a single large pediatric hospital. PT was applied to 168 out of 192 patient with EVD, implementing a presented protocol. Totally 1601 PTs were conducted, and the patients were classified by the diagnosis and the frequency of PTs (1-6x per week). No adverse effects of PTs have been noticed in the cohort, supporting the safety hypothesis. The patients with early mobilization exhibited better outcome compared to bed-resting, in view of shorter stay in hospital, better mobility at discharge and more favorable outcome at discharge, while earlier mobilization and higher PT frequency also have had a positive impact. In conclusion, the study encourages implementation of PT-guided early mobilization of children with EVD in order to improve general outcome.

I have some questions:

1.       The positive impact of PT, where successfully implemented, is undoubted, as the adverse effects in the cohort were unmeasurably rare. However, it is difficult to evaluate the positive contribution of PT out of the context of the primary pathology that lead to the EVD-placement. For example, it is difficult to directly compare the effect of PT in infection, with the effect in malignant or benign neoplasm. Moreover, there is no scoring scale that would enable correlation of the overall severity and the impact of PT on the outcome (i.e. patients that have been ordered bed resting probably have more severe primary condition than these with everyday PT, causing longer stay and worse condition at discharge).

Therefore, is it possible to make correlation of the overall patient condition and the effect of PT more visible in the manuscript?

2. What you suppose that would be age-specific risks (absent in adults), if any?  

Author Response

We are thankful of your thoughtful and constructive comments and questions. We provide responses to your comments and questions below. Thank you for taking the time to review our manuscript.

Does the introduction provide sufficient background and include all relevant references? Can be improved.

            We are grateful for this feedback. In response, we have added further background information to improve reader understanding of this important topic. Newly available references have also been provided to ensure a more current literature base is included to support our work.

Is it possible to make correlation of the overall patient condition and the effect of PT more visible in the manuscript?

 We appreciate this comment regarding the effect of PT on the outcomes based on patient conditions. Ideally, the evaluation of the PT effect within subgroups would require the outcome data on matched patients who did not receive PT; unfortunately, the retrospective nature of this study limited our ability to acquire data on those who did not have PT. In response to your question, we have discussed this as a study limitation.

Line 299: “Comparison of children based on diagnosis who received PT to those who did not were limited due to the lack of known outcomes in children who did not receive PT, as well as comparison of existing research available in this patient population; hence future studies should be performed to corroborate our findings”.

Additionally, there is no accepted scoring scale for overall patient condition. Instead, we have examined the PT outcomes by diagnosis, recommended PT frequency, bedrest order, days between admission and PT evaluation, which are related to patient conditions. In response to your question, we have highlighted these correlational analyses in the statistical analysis section.

Line 133: “We also examined the association between patient conditions and these outcomes; and these conditions included diagnosis, recommended PT frequencies, bedrest orders, and days between admission to PT evaluation.”

What do you suppose would be age-specific risks (absent in adults), if any? 

We thank you for this interesting question. In direct response to your question, we have added associated risks that may be related to the pediatric setting to the introduction. 

Line 62: “Associated risks in the pediatric setting may include escalating and challenging behaviors. Children can sometimes argue, show aggression, or act angry or defiant around others when they are anxious or stressed. A hospital can be a stressful environment and can lead to challenging behaviors inclusive of physical behaviors of pushing/pulling/biting/hitting/kicking, pulling at lines, defiant behaviors, active flight risk, or variable emotions of screaming, yelling, and crying. Children admitted for medical care can often show resistance to medical treatment and necessary care due to past healthcare experiences, temperament, sensory concerns or sensory processing disorders, reactive behaviors due to stressors and pain, mental health diagnoses, and altered mental statuses”.

We also provided a related statement in the discussion as well as the limitations of the manuscript to cover this topic.

            Line 305: “Patient age-specific risks, such as behavior, is likely to be an important factor in rehabilitation outcomes, though difficult to measure and not included in this study”.

Round 2

Reviewer 3 Report

The authors have carefully addressed all my comments and questions. All issues are now clarified, and the manuscript is considerably improved.